# autoFISH: a modular toolbox for sequential single-molecule RNA FISH experiments
Christian Weber[1,2,8], Thomas Defard[1,2,3,4,5,8], Chloé Sturmach [1], Mickael Lelek [2], Hugo Laporte[6,7], Ayan Mallick[6], Maria Isabella Gariboldi[1,2], José-Arturo Londoño-Vallejo [6], Thomas Walter [3,4,5], Charles Fouillade [6], Jacques Bourg [1] & Florian Mueller [1,2,8] ✉

Understanding the role of gene expression in cellular function and tissue organization requires spatial and quantitative detection of individual RNA molecules. Yet, the widespread adoption of automated single-molecule fluorescence in situ hybridization (smFISH) has been limited by the cost of equipment and the complexity of experimental procedures. We present autoFISH, an affordable, user-friendly platform that removes these barriers through open-source hardware components, accessible control software, and integrated analysis tools. The system demonstrates broad applicability by enabling both conventional and signal-amplified smFISH protocols and incorporates an optimized tissue-clearing method that preserves nuclear structures. Testing across multiple cell types and tissue preparations validates the system's reliability and reproducibility, offering a practical solution for scaling spatial transcriptomics research and advancing discoveries in cellular and developmental biology, while significantly reducing costs and the technical expertise required.

Transcription is a fundamental biological process pivotal to development and disease. Individual cells exhibit considerable heterogeneity in their transcriptional profiles, including RNA abundance and intracellular RNA localization. Accurately quantifying the number of RNAs within cells and their locations provides a precise readout of a cell's molecular state and enhances our understanding of RNA regulation. Such measurements require spatially resolved RNA detection in the native cellular environment with sub-cellular accuracy. Here, single-molecule FISH (smFISH) is widely regarded as the gold standard for detecting individual RNA molecules with high sensitivity and precise sub-micrometer localization[1]. Various adaptations of this method have been developed, all based on the principle of targeting RNAs with multiple primary oligonucleotide probes, either directly labeled[2] or indirectly labeled via secondary oligonucleotide probes[3] or signal amplification systems (e.g, branched-DNA[4], HCR-FISH[5], SABER[6]).

RNAs then appear as diffraction-limited spots in fluorescence microscopy images, and their positions can be measured with high spatial accuracy using dedicated analysis workflows[7–9]. When combined with cell segmentation[10], the RNA abundance in each cell can be estimated, and statistical analysis can be used to quantitatively describe subcellular RNA localization[11,12]. Crucially, smFISH preserves the spatial context of cells while

revealing their transcriptional states. This dual output facilitates advanced quantitative analysis of tissue architecture by integrating molecular profiles, cellular identities, and spatial information. Cell types can be identified using standard scRNA-seq clustering approaches[13], and their annotation can be further refined by incorporating spatial data[14]. This enables large-scale investigation of features such as cell-cell communication, tissue composition, and complex tissue gradients (Palla et al.[15]).

A significant limitation of the original smFISH methods is that the number of spectrally distinguishable fluorophores constrains the number of distinct RNA species that can be visualized simultaneously. To overcome this limitation, new methods have been developed that rely on iterative hybridizations, i.e., alternating cycles of probe hybridization, imaging, and signal removal[16–18]. A general feature of these approaches is that the primary probes carry one or more readout sequences that can be detected by a secondary, complementary readout probe. Usually, primary probes against all RNA targets are hybridized on the bench, while the readout probes are hybridized directly on the microscope with an automated fluidic system. Signal removal can be achieved by various methods; the most commonly used are stripping with high-stringency buffers[19], fluorophore removal via chemical cleavage[20], or displacement with a tertiary stripping probe[21]. In sequential smFISH experiments, the number of targeted species is linearly

[1]Institut Pasteur, Université Paris Cité, Photonic Bio-Imaging, Centre de Ressources et Recherches Technologiques (UTechS-PBIC2RT), Paris, France. [2]Institut Pasteur, Imaging and Modeling Unit, Université Paris Cité, Paris, France. [3]Centre for Computational Biology (CBIO), Mines Paris, PSL University, Paris, France. [4]Institut Curie, PSL University, Paris, France. [5]INSERM, U900, Paris, France. [6]Institut Curie, Inserm U1021-CNRS UMR 3347, University Paris-Saclay, PSL Research University, Centre Universitaire, Orsay, France. [7]Institute of Cell Biology (Cancer Research), University Hospital Essen, Essen, Germany. [8]These authors contributed equally: Christian Weber, Thomas Defard, Florian Mueller. ✉e-mail: fmueller@pasteur.fr

correlated with the number of iterations and the number of fluorophores used. Multiplexed barcoding approaches have driven the field of spatial transcriptomics by dramatically increasing RNA detection throughput[22]. Here, each RNA species carries several readout sequences forming unique RNA-specific barcodes. Although readout sequences are reused across RNAs, each hybridization round visualizes a different set of genes. RNA identities must be decoded post-analysis. When combined with error correction, this combinatorial method enables the detection of hundreds to thousands of RNA species.

While several commercial systems offer convenient turnkey solutions with increasingly large panel sizes, they are costly in terms of instrumentation and reagents and lack flexibility. Exploratory analyses, time courses, dose-response studies, and replicates thus demand substantial financial resources. While large RNA panels are often necessary, not all studies require such depth; smaller, more flexible panels may suffice. For example, a small probe panel with 10 genes can be adequate to resolve the main cell types in tumor cryosections[23].

While these commercial methods are user-friendly, they often lack flexibility across microscopy (e.g., imaging modalities) and experimental protocols (e.g., RNA detection approaches). This limitation can be critical

for challenging samples, such as tissue sections, where weak RNA signals benefit from amplification to improve detection reliability. Several widely used amplification strategies include branched DNA[4,24] (Wan et al.; Xia et al.), hybridization chain reaction (HCR)[5,25], and SABER[6]. SABER is a compelling choice, pairing a simple base protocol with scalable amplification levels.

Taken together, home-built solutions remain a crucial option for addressing many research questions. To address this need, we propose autoFISH, an end-to-end open-source workflow comprising hardware, software, and experimental protocols for automated smFISH experiments designed for flexibility and affordability. The autoFISH toolbox relies on a versatile and affordable fluidic system, for which we provide detailed building plans. Flexibility comes in several flavors. First, the imaging modality can be chosen depending on the desired application, as the fluidic system required for sequential hybridizations can be mounted on various microscopes. Second, protocols can be readily modified to meet experimental needs, such as by incorporating signal amplification or tissue clearing. Lastly, smaller-scale experiments can be readily performed because probe design and synthesis are amenable to control.

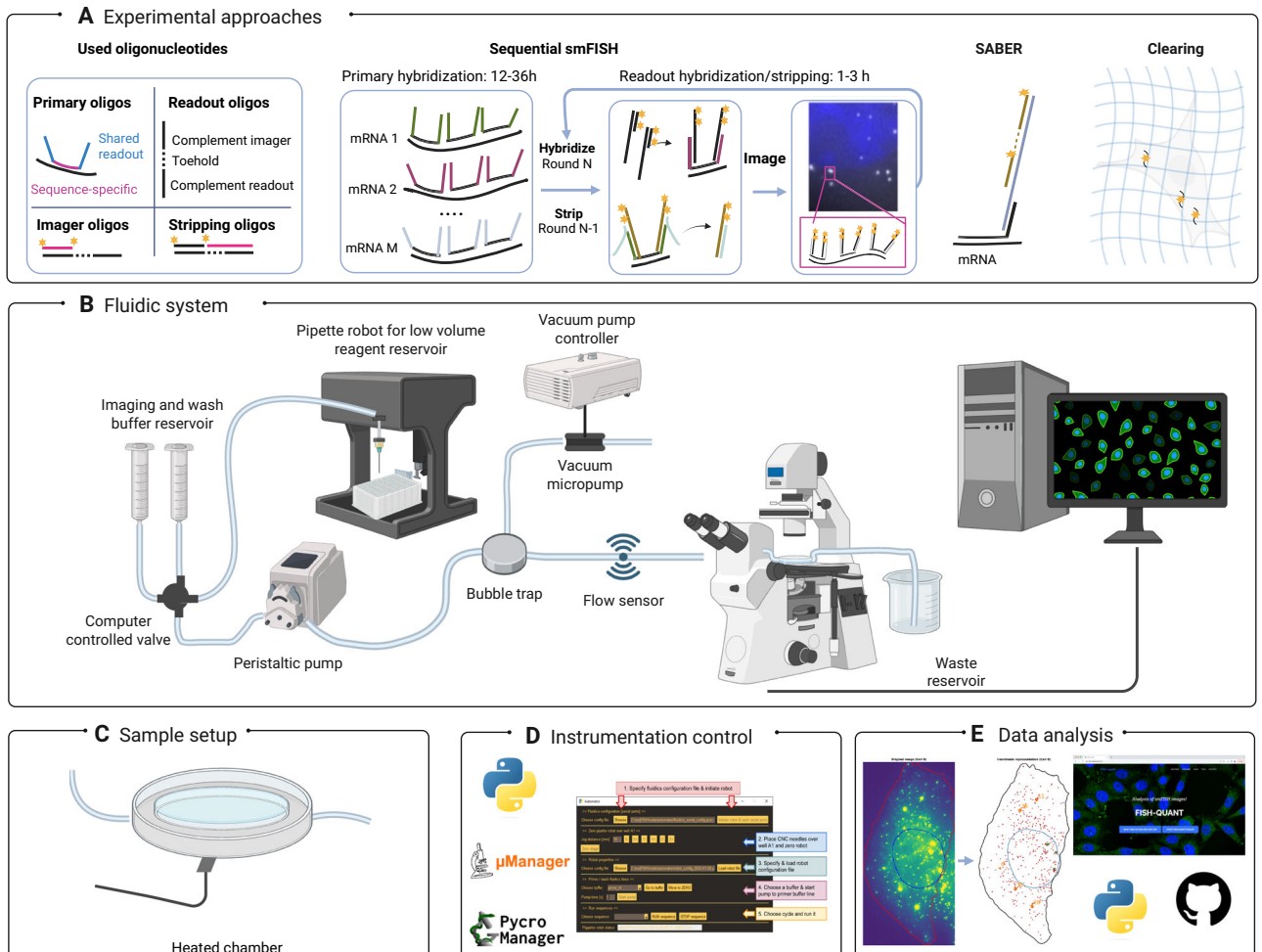

**Fig. 1 | Summary of the autoFISH workflow, from wetlab to analysis. A** Schematic of the experimental workflow. Used oligonucleotides: Overview of primary, readout, imager, and stripping oligos and their nomenclature. Sequential FISH: Primary oligos are hybridized once; across N rounds, readout oligos reveal defined RNA subsets, while stripping oligos remove signals from previous rounds N-1. SABER: Signal amplification is achieved via DNA concatemers, providing multiple binding sites for fluorescent imagers. Clearing: RNA targets are anchored to a polymer, and the sample is cleared to reduce autofluorescence and nonspecific background. **B** Overview of the fluidics system with the different components. **C** Samples can be placed in an imaging chamber for high-quality microscopy with fluidic inlet/outlet ports. **D** The Python-based control software coordinates reagent delivery from the fluidic system with the microscope's acquisition sequences. **E** Provided software enables the analysis of different hybridization rounds and yields RNA abundance measurements for each segmented cell. Created in BioRender. MULLER, F. (2026) https://BioRender.com/psuzxy7.

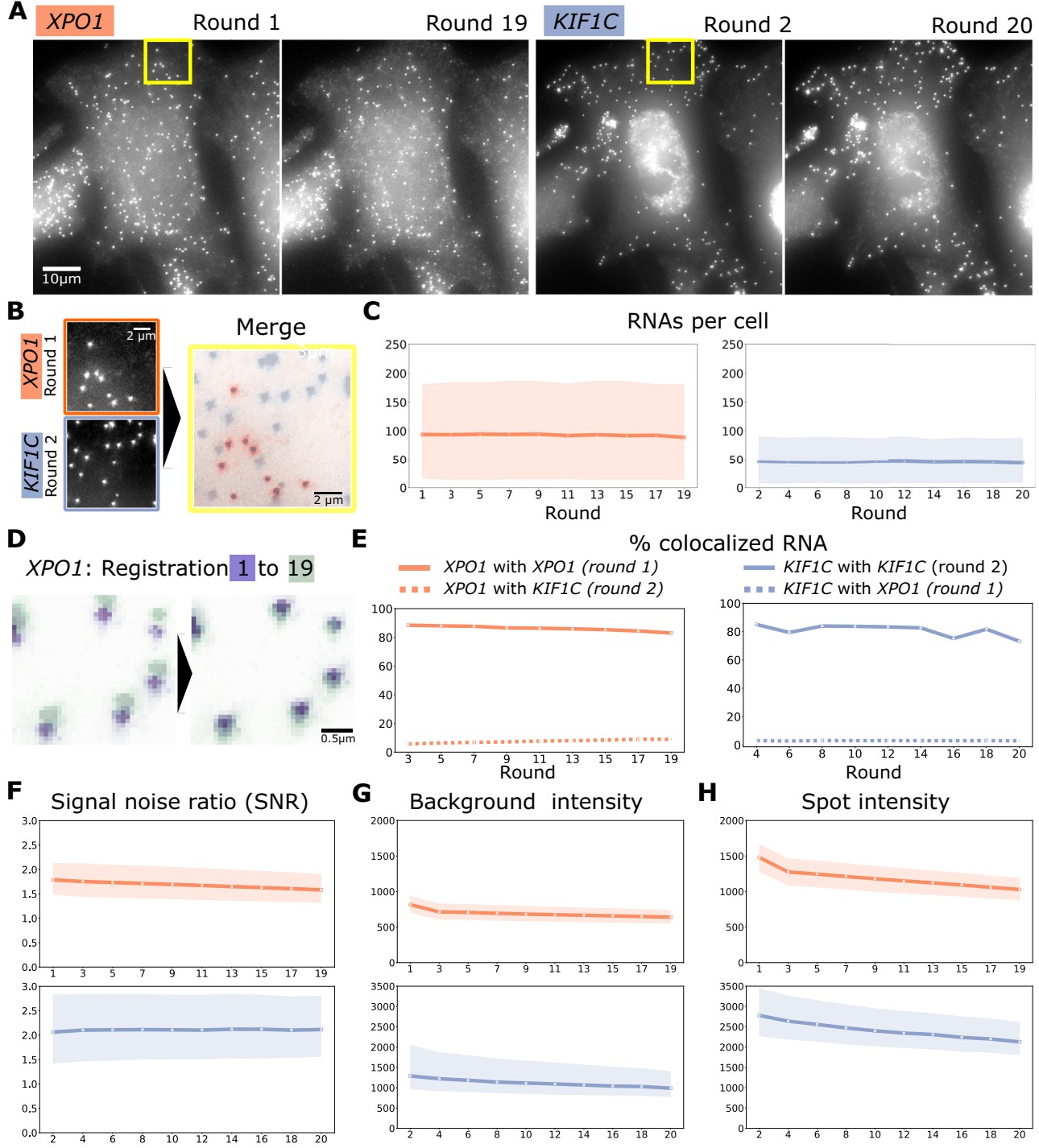

**Fig. 2 | Robustness of autoFISH over 20 sequential hybridization rounds.**
**A** Images of the same HeLa cell with either *XPO1* and *KIF1C* for the respective first and last hybridization rounds. Please note that the nuclear background observed in *KIF1C* is due to nonspecific binding of some primary probes. The same intensity scaling [min, max] was used for images of each gene: *XPO1* [374, 2216], *KIF1C* [2334, 5708]. **B** Zooms in (yellow rectangles in A) showing *XPO1* (round 1) and *KIF1C* (round 2). **C** RNA counts per cell across rounds for *XPO1* and *KIF1C*.

**D** Example of image registration with images from round 1 (purple) and round 19 (green). **E** Percentage of colocalized RNAs with reference to the first respective hybridization round (Distance threshold for co-localization = 0.5 μm). **F–H** Mean signal-to-noise-ratio (SNR), background intensity, and spots intensity across rounds. Number of field of views = 25; number of cells = 453; number of RNAs: KIF1C = 305 K, XPO1 = 55 K. In panels C,F,G,H lines represent the median; shaded areas indicate the interquartile range (IQR, 25th–75th percentiles).

We demonstrate the applicability of autoFISH on two experimental datasets. Firstly, HeLa cells, which we use to benchmark the performance of the protocol and illustrate its flexibility by alternating between standard and amplified FISH experiments with SABER[6], and secondly, mouse lung tissue, which we use to visualize several marker genes from our earlier study[26]. Here, we introduce a modified tissue-clearing protocol that improves nuclear signal. In summary, we provide a comprehensive toolbox for establishing automated, sequential smFISH experiments at a lower cost.

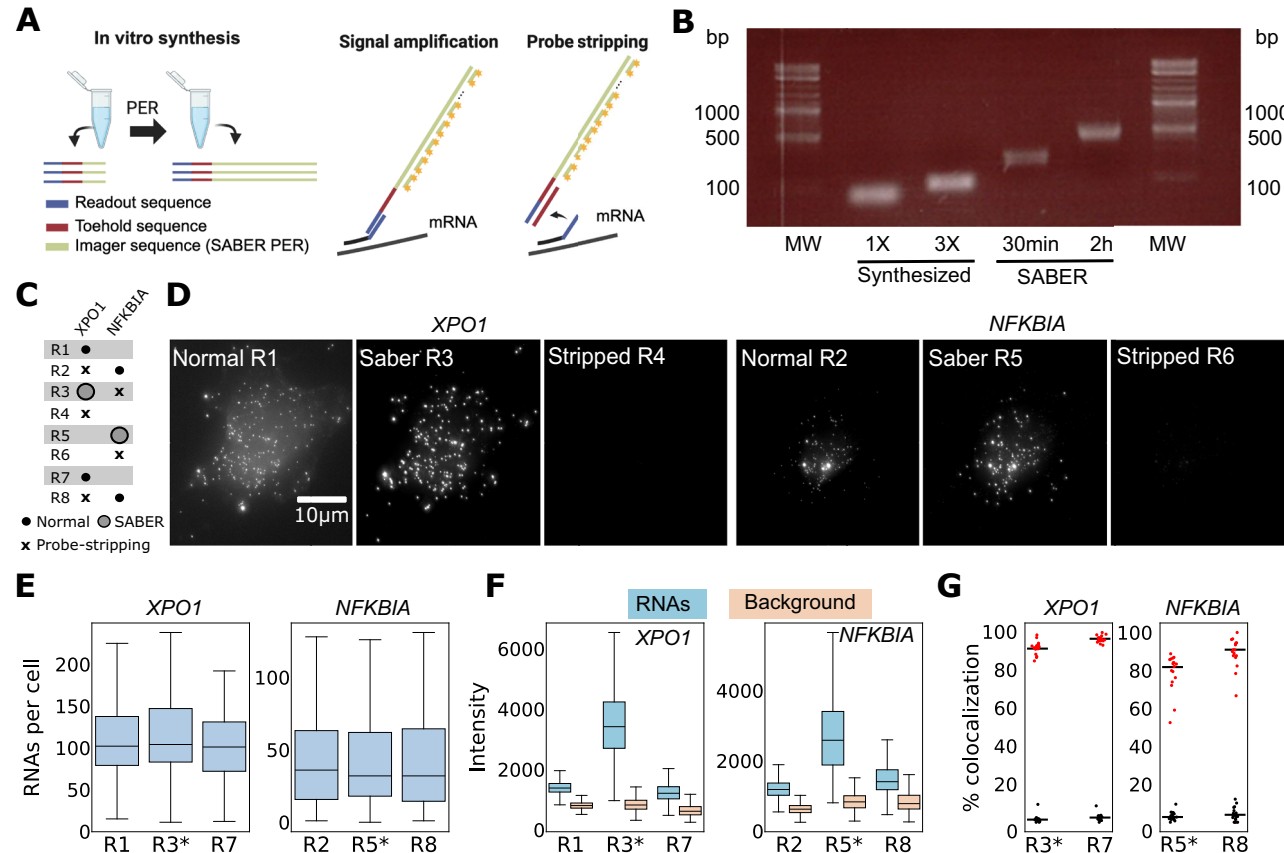

**Fig. 3 | Signal amplification with SABER across hybridization rounds. A** SABER amplifies a specific sequence serving as a binding region for several imaging probes. These oligos also contains a toehold region (red) that permits probe stripping. Created in BioRender. MULLER, F. (2026) https://BioRender.com/akavdgj. **B** 2% agarose gel electrophoresis shows readout oligos with either 1 or 3 repeats, or SABER amplification with two durations (30 min and 2 h). Gel was visualized using a Fisher Bioblock Scientific UV imaging system. An image of the entire gel was captured on thermal paper and subsequently digitized and is shown with minimal cropping. **C** Hybridization conditions for the eight rounds (R1-R8): small, filled dots indicate a standard smFISH; larger dots indicate SABER FISH; x indicates stripping of the previous round. **D** Images of the same HeLa cell across different hybridization rounds. Images for the same gene are shown with the same intensity scaling. **E** RNA levels per cell across rounds. SABER runs are indicated with an asterisk. **F** Quantification of intensities of RNA and background across rounds. **G** Gene colocalization across imaging rounds (Distance threshold for co-localization = 0.2 µm). Colocalization of *XPO1* and *NFKBIA*. Spots were compared to reference detections from round 1 (XPO1) and round 2 (*NFKBIA*). Red: same-gene colocalization; Black: cross-gene colocalization. Small points indicate colocalization coefficients for each field of view; bars show the pooled results across all fields. Number of fields of view = 25; number of cells = 58; number of RNAs: *NFKBIA* = 2.6 K, *XPO1* = 6.5 K. Boxplots: central line = median; box = Interquartile range (Q1–Q3); whiskers = max/min data points within 1.5× IQR. Outliers excluded.

## Results

Here, we present autoFISH (Fig. 1), a universal toolbox for performing sequential smFISH, with instrumentation that can be built in-house at an affordable cost and adapted to specific needs and experimental protocols using standard buffers. autoFISH is based on several previous studies[3,6,8,27,28] and is entirely open-source. We also provide Python workflows to analyze these data and infer single-cell RNA quantification. autoFISH is hosted on GitHub (see Methods) and includes detailed build plans, control software, and experimental protocols. To validate and demonstrate the capability of autoFISH, we implemented and validated different protocols, including workflows for signal amplification and tissue clearing on both cultured cells and tissue sections.

### Computer-controlled fluidic system for automated experiments

A sequential smFISH experiment consists of several key steps: hybridization of primaryprobes overnight at 37 °C, sample mounting on the microscope, and alternating imaging and incubation steps for fluorescently labeled secondary probe hybridization and displacement of secondary probes from the previous round (Fig. 1A).

The core of autoFISH is a computer-controlled fluidic system that performs the sequential hybridization steps and automates imaging (Fig. 1B). This system enables flushing buffers from syringes, providing either large buffer reservoirs or single-use buffers stored in multiwell plates, which can be accessed via a pipetting robot. A computer-controlled valve allows buffer selection, and flow control is achieved via a peristaltic pump, with flow rates monitored by a flow sensor. Lastly, air bubbles are removed from the buffers with a bubble trap.

Samples can be mounted in an imaging chamber, which is ideal for single-condition cell culture or tissue samples (Fig. 1C). We emphasize that placing the pump immediately after the valve yielded better performance than placing it at the end of the fluidic system (after the imaging chamber). Based on our experience, this resulted in higher bubble-trap efficiency and reduced deformation of the imaging chamber during pump action, which can otherwise lead to loss of microscope autofocus.

We implemented a modular, open-source Python package that controls all components of the fluidic setup to perform automated buffer exchanges (Fig. 1D). The code's modularity facilitates adding new components from other providers, such as a different pump. The user can flexibly define fluidic runs in a human-readable text file that contains all buffers, pump durations, and incubation times, as well as optional steps, such as performing DAPI staining only for one run or adding hybridization steps, e.g., for signal amplification with SABER.

For fully automated experiments, imaging must be automated as well. As a design choice, we do not directly control microscopes within our code;

instead, we provide methods to interface with acquisition software packages. Currently, we support acquisition in Micromanager via Pycromanager, or triggered acquisition via a TTL pulse or a shared synchronization text file (see Methods for details). With these solutions, autoFISH can already be used with various imaging systems, and new trigger approaches can be added as needed.

We provide a graphical user interface (GUI) to facilitate usage. This interface can be used to control the entire workflow: load the configuration files, initiate the fluidic system, trigger the microscope, and then launch either an individual fluidic run or a fully automated run, in which imaging and fluidic runs are performed sequentially and iteratively.

## autoFISH with 20 hybridization rounds

We first validated the robustness of autoFISH by assessing its ability to detect RNA and remove signals. In sequential smFISH, RNA detectability is determined by several key performance metrics. Probe stripping efficiency is fundamental and is evaluated by comparing images of the same target RNA before and after stripping. The absence of residual signal in post-stripping rounds is critical to prevent false positives in subsequent hybridization cycles. Furthermore, consistency in spot counts across identical rounds serves as a crucial metric; a stable number of detected RNA spots indicates reproducible detection efficiency throughout the detection rounds. This is complemented by cross-round colocalization analysis, which directly assesses whether the same individual transcripts are reliably identified across cycles. A high colocalization rate (e.g., ~90%) confirms the consistent detection of the same RNA target. To ensure the technical quality of the images, we monitored the signal-to-noise ratio (SNR), defined as the spot intensity relative to mean local background fluctuations. While a stable SNR is essential for reliable automated detection, tracking absolute signal and background intensities provides more detailed insight. For instance, a proportional decrease in both RNA signal and background intensity—resulting in a stable SNR—suggests a minor reduction in probe binding without compromising overall detectability. Collectively, these metrics allow evaluation of whether the system maintains robust detection across sequential hybridization rounds.

To validate our approach, we performed a 20-round experiment in which we alternated the detection of two genes with distinct subcellular localizations[11]: *XPO1* (randomly distributed) and *KIF1C* (localized to cell protrusions). We consistently detected the same RNA species across successive rounds (Fig. 2A). Furthermore, the absence of residual signal following stripping confirms the efficiency of probe removal (Fig. 2B). Quantitative analysis revealed a stable number of detected spots per round (Fig. 2C) with high spatial conservation; approximately 90% of spots colocalized across rounds (Fig. 2D, E and Supplementary Fig. 1), matching benchmarks for dual-color RNA labeling[3]. While the SNR remained constant throughout the experiment (Fig. 2F), closer inspection indicated that successive washing cycles slightly reduced both the RNA spot intensity and the background (Fig. 2G, H). Collectively, these results demonstrate robust, reproducible RNA detection over 20 hybridization rounds.

## Flexible signal amplification

In standard smFISH, each RNA molecule is targeted by a set of primary probes, resulting in a locally concentrated fluorescent signal at the transcript location. To enhance detection efficiency—particularly in samples with high autofluorescence or low RNA signal—autoFISH employs a multi-layer signal-amplification strategy.

In the basic autoFISH workflow, signal amplification is achieved through a modular, three-component assembly based on previous work[27]. Primary probes are synthesized with distinct readout sequences at both the 3' and 5' ends, which serve as binding sites for intermediate readout probes during detection. To maximize signal while maintaining cost-efficiency, imaging probes—each conjugated with two fluorophores—are pre-hybridized to these readout probes. This configuration enables the recruitment of up to four fluorophores per primary probe, enhancing the SNR. While brightness can be further increased by directly labeling the intermediate readout probes, the standard configuration typically provides sufficient sensitivity while minimizing reagent costs.

SABER offers a robust, programmable alternative[6] by in vitro extension of oligonucleotides with repeated sequences before use in hybridization (Fig. 3A). These extended probes can then be bound by multiple imaging probes, with signal intensity precisely controlled by the amplification reaction duration (Fig 3B). A key advantage of this pre-hybridization amplification is that it does not extend the actual smFISH experiment. To simplify this approach, we also evaluated synthetized oligonucleotides with three repeat binding sites—the practical limit for a ~150 nt synthesis. We further enhanced signal strength by labeling imaging probes at both the 3' and 5' ends. While signal intensity scaled with the number of binding sites, dual-end labeling yielded less than a twofold increase, likely due to steric hindrance or fluorophore quenching (Supplementary Fig. 1). These results suggest that for moderate amplification, synthetized probes with three repeated binding sites for the imaging probes provide sufficient signal enhancement.

We integrated SABER into autoFISH by replacing the imager sequence on the readouts with the SABER PER sequence required for amplification (Fig. 3A). We then optimized the protocol to be compatible with the fluidic system. In short, we replaced the formamide from the original buffers with the less toxic Ethylene Carbonate (EC). We also hybridized the SABER probes in two steps on the fluidic system: first, the amplicons, then the imaging probes. We performed the hybridization of the amplicons in a more stringent buffer, resulting in a substantial decrease in background. We also found that adding even a moderate amount of dextran (5% was compatible with our fluidic system) reduced the amount of imaging probes required, thereby reducing experimental costs. With these changes, SABER can be performed on the fluidic system. Interestingly, the flexibility in the fluidic control allows us to perform SABER hybridization, which requires an additional hybridization step, in selected runs only, as demonstrated in the validation experiments.

We then performed an experiment targeting two genes (*XPO1* and *NFKBIA)*, alternating between normal and SABER hybridization, and included a control round in which we only stripped SABER without a new hybridization (Fig. 3C). Qualitative inspection of the images shows clear amplification of SABER for both genes, with the same RNAs visible and SABER resulting in increased signal intensity (Fig. 3D). Importantly, the SABER amplicon can also be efficiently stripped and the signal removed (Fig. 3D). A more detailed analysis confirmed this qualitative impression, showing a constant amount of detected RNA per round (Fig. 3E). Analysis revealed that SABER substantially increased RNA signal intensity, albeit with a marginal increase in background levels (Fig. 3F). Importantly, co-localization between detection rounds (both standard and SABER) for the same gene is high (Fig. 3G). In contrast, co-localization between the two genes is very low, indicating no cross-detection.

These experiments demonstrate the integration of SABER into the autoFISH workflow to amplify RNA signal intensity. Furthermore, they highlight the control software's flexibility, which allows adjustment of hybridization conditions across specific imaging runs. This combined approach enables high-sensitivity detection while maintaining the automated throughput required for complex multiplexing cycles.

## autoFISH on cleared mouse lung tissue

In a recent study, we performed smFISH on mouse lung tissue to detect RNA marker genes for different cell types[26]. While the imaging quality was sufficient, we encountered substantial background fluorescence. It has been shown that this background comes from both tissue autofluorescence and nonspecific probe binding to various cellular structures such as proteins and lipids[29]. To mitigate this, tissue-clearing methods can be used to remove these interfering components. In these approaches, samples are embedded into a nonswellable polyacrylamide hydrogel (PA) where RNAs are covalently anchored (Fig. 4A). Several anchoring strategies exist, including (i) the use of acrydite-modified oligonucleotide (dT) probes, which hybridize to mRNA poly(A) tails and covalently integrate into the polyacrylamide

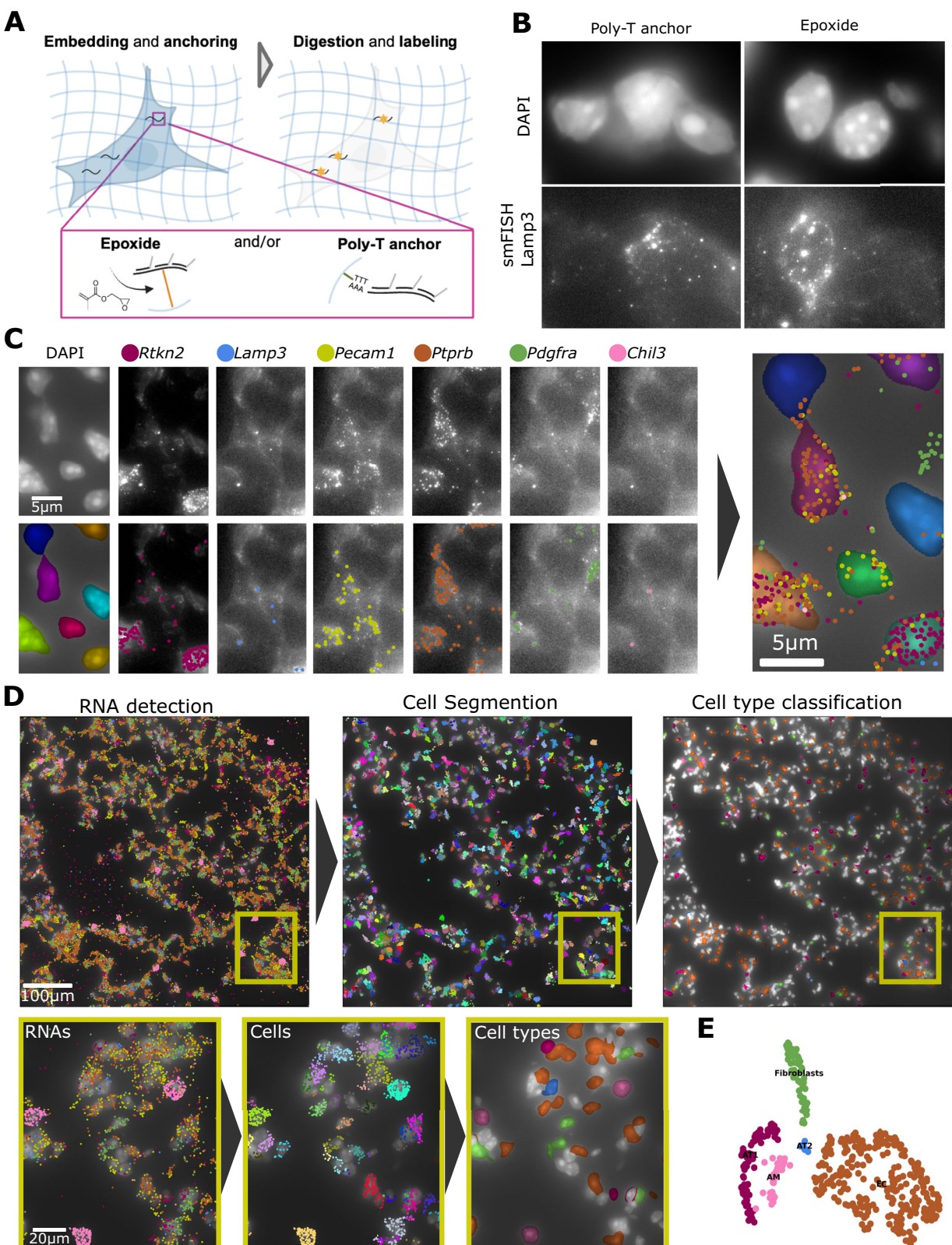

**Fig. 4 | Clearing reduces background and allows RNA detection in tissue.**
**A** Principle of clearing. RNAs can be anchored with epoxide and/or Poly-T oligos carrying an acrydite moiety. Created in BioRender. MULLER, F. (2026) https://BioRender.com/gs5vmrv. **B** Influence of epoxide on nuclear signals. DAPI and smFISH images after clearing with Poly-T anchoring and epoxide. **C** Nucleus segmentation and RNA spots detection across rounds. **D** Results of analysis workflow from RNA detection to cell segmentation and cell type classification. The analysis

was performed across 6 3D fields of view, resulting in the segmentation of 1.1 K nuclei. RNA counts per gene: *Rtkn2* (6.5 K), *Lamp3* (1.0 K), *Pecam1* (7.4 K), *Ptprb* (12.8 K), *Pdgfra* (3.8 K), and *Chil3* (2.6 K). The yellow rectangle shows a region provided with a zoom-in. **E** Louvain clustering of the spatially resolved RNA profiles. Cell-type mapping based on our reference scRNAseq data by matching the cosine distance between cluster centroids.

network[29], and (ii) chemical RNA anchoring agents such as LabelX[30] or MelphaX[31]. The embedded sample can then be treated with Proteinase K and SDS to remove proteins and lipids while retaining the anchored RNAs, resulting in a significantly reduced background.

When applying the protocol with the Poly-T anchoring probes to lung tissue, the background was reduced, and RNAs could be detected. However, we found that the nuclear signal was often less distinct than in samples anchored with epoxide (Fig. 4B). For samples such as lung tissue with a locally high nuclear density, this could adversely affect nuclear segmentation. We hypothesized that this could be due to reduced chromatin constraints after digestion. In a recent study, an epoxide (GMA) was introduced as a universal linking agent in Expansion Microscopy (ExM) for proteins and RNA[32]. GMA possesses two reactive sites, allowing covalent linkage of various functional groups—particularly amine moieties—to the acrylamide gel matrix. Since the polymers used for clearing and ExM are similar, we speculated that GMA could also be used in clearing protocols and serve as a structural scaffold.

We compared DAPI staining of HeLa cells after clearing under three conditions: no anchoring, anchoring with GMA, and anchoring with GMA plus poly-T probes. When GMA was used—either alone or with poly-T probes—nuclear staining appeared crisper (Fig. 4B). To provide a more comprehensive comparison, we analyzed these images using a quantitative sharpness metric (see Methods). This analysis confirmed our qualitative observations and demonstrated increased sharpness of the DAPI signal upon addition of GMA (Supplementary Fig. 2).

We then performed autoFISH in cleared mouse lung tissue against six mRNA targets from our previous study[26]. Individual RNA molecules for each target were successfully detected (Fig. 4C). Because dedicated membrane markers were not used, we applied our analysis package, ComSeg[33], to segment cells using nuclear staining and to detect RNA point clouds (Fig. 4D). The resulting gene expression profiles then enable clustering (Fig. 4E) and mapping the identified cell types back onto the image, revealing their spatial locations.

In summary, we established an optimized tissue-clearing protocol incorporating GMA to improve retention of nuclear signals. We validated this method in mouse tissue, demonstrating robust detection across a panel of marker genes.

## Discussion

smFISH is a powerful technique for visualizing individual RNAs within their native cellular context. By performing multiple cycles of probe hybridization, imaging, and probe removal, the number of RNA species that can be analyzed is greatly increased. However, this also requires more complex protocols and specialized equipment. Although commercial solutions are available, they are costly and lack flexibility. Here, we introduce autoFISH, a modular toolkit for automated, sequential smFISH that leverages validated, optimized workflows.

In this work, we demonstrate the performance of autoFISH across cell lines and tissues, including up to 20 hybridization runs. While the throughput of this method is lower than that of commercial multiplexed solutions, it offers some advantages. First, because each RNA is imaged individually, spatial crowding is not a concern, thereby allowing even highly expressed genes to be targeted. Additionally, by ordering probe sets for each gene individually, we can customize the selection of targeted genes. One of the main strengths of autoFISH is its flexibility. We demonstrate this by applying various experimental protocols, showcasing flexible control over fluidic runs via optional steps, and by implementing different methods to trigger acquisitions on the microscope.

For more challenging samples, we provide experimental protocols for signal amplification and tissue clearing that are compatible with the autoFISH fluidic system. We illustrate how signal amplification can be achieved with SABER and demonstrate that it can be performed on specific runs via the flexibility of the fluidic control. The proposed use of epoxide in tissue clearing enhances preservation of nuclear signals, promising to improve segmentation in densely packed samples. Additionally, the epoxide used

(GMA) is inexpensive, allowing for cost-effective experiments. Finally, epoxide anchoring is more versatile than targeting poly(A) tails, which limits its application to mature mRNAs. This versatility suggests that other RNA species, such as nascent or long non-coding RNAs, could potentially be anchored using this method, thereby reducing the likelihood of their loss during the clearing step.

The autoFISH control software features an open, modular architecture designed for integration with diverse imaging systems. Implemented in Python, the software provides a flexible framework for integrating various acquisition triggers and hardware components, such as valves and pumps. This open-source approach facilitates the future integration of Python-based libraries for real-time quality control and data processing.

In summary, autoFISH significantly lowers the barrier to entry for research groups implementing sequential smFISH. While commercial platforms excel at large-scale, high-multiplexing discovery experiments, in-house solutions - such as the one presented here - provide a cost-effective complement. By utilizing high-throughput data e.g. scRNA-seq or large-scale spatial screens , to identify refined gene signatures, researchers can use autoFISH to probe specific targets at a substantially reduced cost. Finally, as an open-source platform, autoFISH is designed for community-driven expansion, such as the development of new modules and the integration of diverse experimental protocols.

## Methods

This section provides an overview of the methods. For detailed, up-to-date information, readers are referred to the following resources: https://github.com/fish-quant/autofish (control code), https://doi.org/10.5281/zenodo.17965896 (fluidic building plans and experimental protocols), and https://github.com/fish-quant/autofish-analysis (analysis code).

### Instrumentation and control

autoFISH relies on an automated fluidic system for buffer exchanges. Python control software was developed to operate the fluidic system and communicate with the microscope to initiate acquisitions. autoFISH is implemented as a lightweight Python package with minimal dependencies. It utilizes PySerial for communication with various hardware components, and its modular design enables the easy addition of new components, such as pumps or valves from different suppliers. autoFISH can be controlled with a graphical user interface.

### Microscopy

autoFISH does not allow direct control of microscopes but provides several options for triggering image acquisition. This makes autoFISH highly flexible, as it can be used across various microscopes with minimal adaptation. At the time of publication, the following options are available, but new ones can be easily added:

Pycromanager[34] is suitable for microscopes controlled by Micro-manager. The user can define a settings file with the relevant acquisition parameters and a position list. autoFISH will then create a multi-D acquisition event that enables imaging after each hybridization round.

Synchronization with a text file: either the file's content (e.g., a switch from 0 to 1) or its existence can be used to start an acquisition. Once the acquisition is complete, the file content can be set back to 0, or the file can be deleted, signaling to autoFISH that the next hybridization round can begin.

Communication with TTL trigger: This is suitable for microscopes that integrate incoming and outgoing TTL trigger events into their acquisition routines. In autoFISH, we use an Arduino to send and receive TTL triggers: a TTL trigger initiates an acquisition, and an incoming TTL trigger signals that the acquisition is terminated and that the next hybridization rounds can begin.

### Fluidic system

We use a custom-built fluidic system that combines elements from published approaches (merFISH[20] and ORCA[27]). Detailed building plans and part lists are on GitHub (https://doi.org/10.5281/zenodo.17965896). The

system consists of the following main components, which are widely available and affordable: a peristaltic pump to accurately control flow rate, a bubble trap connected to a vacuum pump to remove air bubbles, and a flow sensor to monitor flow and detect potential issues, such as obstructed fluidic lines or large air bubbles.

Buffers are stored either in deep-well 96-well plates (each well holding up to 2 mL) or in syringes. The former usually contains hybridization buffers used only once, and the syringes contain general-purpose buffers for washing and imaging. The 96-well plates are placed on a pipette robot built from a CNC milling machine. These robots can be readily adapted by incorporating a 3D-printed insert into the spindle holder to accommodate a needle connected to a fluidic line. The needle can aspirate liquid from the multi-well plates. A computer-controlled valve selects buffers from different inlets from the fluidic lines connected to the pipette robot or the syringes.

### Sequential smFISH: instrumentation and microscopy

Detailed experimental protocols are available at https://doi.org/10.5281/zenodo.17965896.

In experiments on the fluidic system, we used the less-toxic Ethylene Carbonate (EC) instead of formamide, as previously described[29]. In most previous studies, hybridizations on the fluidic system were done without dextran. For some experiments, such as signal amplification with SABER, we found that even low concentrations of dextran help reduce oligo concentrations, improve signal, and decrease background. We found that, with our peristaltic pump (Ismatec Reglo Digital), buffers containing up to 5% dextran can be used without detrimental effects on flow rates. It is essential to filter all buffers (using 0.22 μm filters) to prevent clogging of the fluidic lines.

Two different microscopy systems were used in this study. Data for Figs. 1, 2, and 4 were acquired on a Nikon Ti Eclipse equipped with a Lumencor Spectra X LED light source, a 60× 1.4 NA objective, and a Prime BSI Express sCMOS camera, and dedicated monoband filters from Semrock (DAPI-5060C-OMF, Cy3-4040D-000, Cy5-4040D-000). The system was controlled via Micro-Manager, with automated acquisition through Pycro-Manager in autoFISH. Data for Fig. 3, and Supplementary Figs. 1 and 2 were acquired on a Leica Thunder (DMi8) with a Leica K8 camera, 613×1.4 NA objective, Leica LED8 light source, Leica quad-band DFT51011 filter set with a motorized emission wheel containing 460/80, 535/70, 590/50, 642/80, and 100% open positions. The system was operated using Leica LAS X software with the trigger module, and automated acquisition was achieved via TTL signals exchanged between an Arduino and the microscope through the autoFISH program.

### Sequential smFISH: probe design and signal amplification

In this study, we employed the labeling strategy described by Bintu et al[27], which uses four oligos (Fig. 1): primary oligos, readout oligos, imager oligos, and displacement oligos. For each gene, 20–50 primary oligos are designed, consisting of two parts. The first part is complementary to a portion of the target gene's RNA. The other part is an identical readout sequence placed on 3' and 5' of each oligo[27]. All oligos targeting the same RNA share this readout sequence. Note that these primary oligos are not fluorescently labeled. A readout oligo with the complementary readout sequence is then used to detect a specific RNA. This oligo is pre-hybridized with an imager oligo carrying two fluorophores. Each primary oligo is thus labeled with up to 4 fluorophores. This design is cost-effective because the same imager oligo can be used with different readout oligos. A third region on the readout oligo— the toehold—can be used to remove the fluorescent signal. Using a displacement oligo complementary to the toehold and readout sequence, the readout oligo can be efficiently stripped off, and the resulting oligo duplex can be washed out. In an experiment, all primary oligos are hybridized simultaneously during a primary incubation step, while readout oligos are hybridized sequentially to image each RNA species. Multiple RNA species can be imaged simultaneously using imager oligos conjugated to spectrally distinct fluorophores. Interestingly, displacement oligos to remove readout oligos from previous rounds can be hybridized simultaneously with new readout oligos. This allows for faster experiments.

We observed that some of the original readout sequences yielded lower signal intensities in RNA smFISH. We developed a simple strategy to systematically test their quality (Supplementary Fig. 3). In short, we used a probe set against a housekeeping gene (XPO1) carrying a validated readout sequence (RO1). We then design an intermediate oligo with a 5' region complementary to RO1 and a 3' region complementary to the readout to be tested. The to-be-tested readout can then be hybridized with an imager oligo and visualized. This approach eliminates the need to synthesize a reference probe-set for each new readout, but requires only a simple adaptor oligo. For an up-to-date list of validated readout and imager sequences, please refer to https://doi.org/10.5281/zenodo.17965896.

SABER provides a simple, programmable method for signal amplification[6]. We successfully achieve up to a 10-fold amplification without inducing large oligo aggregations. Specific hybridization rounds can be amplified and stripped by using the amplified SABER sequences instead of the imager sequences on the readout oligos. During SABER optimization, we observed abundant nonspecific dots arising from either nonspecific primary probes or amplicons, likely due to signal amplification. To reduce this background, we found that a 3-step hybridization protocol performs best when stringency is optimized at each step. First, primary probes are hybridized on the bench, using a more stringent buffer (1X SSC, 30% formamide, 10% dextran) rather than our standard buffer (2X SSC, 20% formamide). Second, the amplicons are hybridized in the fluidic system for each round using 1xSSC, 20% EC, and 5% dextran. Lastly, the imaging oligos are hybridized in the fluidic system (2X SSC, 10% EC, 5% dextran). The melting temperature of these oligos is relatively low because they contain no Guanine, and a more stringent buffer results in a loss of signal intensity. The low dextran concentration does not affect flow rates, allowing a reduction in the concentration of the imaging oligos.

Target-specific oligos can be designed using various computational frameworks[35]. For this study, we utilized Oligostan[3], a tool we developed specifically for the bioinformatic design of smFISH probes. All probe sequences are available as Supplementary Data 1.

The choice of synthesis method is typically dictated by the required probe scale. For small-scale applications involving several hundred oligos, we utilize pre-synthesized oligo pools (e.g., IDT oPools) at a concentration of 50 pmol per oligo. These pools do not require further amplification and provide sufficient material for several hundred hybridization rounds on 12 mm coverslips. This approach enables modular assembly of custom hybridization panels tailored to specific biological questions. For larger-scale experiments requiring higher probe counts, we employ complex oligo pools that necessitate enzymatic amplification prior to use[28].

### Clearing

Sample clearing reduces background signal in smFISH by removing tissue autofluorescence and reducing probe off-target binding to lipids and proteins[29]. Briefly, RNAs are first anchored to a polyacrylamide matrix using poly (A)-targeting oligos and/or an epoxide (GMA). Once the matrix has solidified, tissues are incubated overnight at 37 °C in a Proteinase K solution (20 mg/mL). The next day, tissues are washed, and hybridization of readout oligos is performed.

### Sequential smFISH in HeLa cells

To validate the autoFISH workflow, HeLa cells (ATCC CCL-2) were cultured in DMEM (Gibco 21063-029), supplemented with 10% FBS and 1% penicillin-streptomycin at 37 °C with 5% $CO_2$. Cells were then fixed in 4% paraformaldehyde (PFA) and permeabilized with 70% ethanol at −20 °C overnight. After rehydration, the samples were hybridized overnight with primary probes against *XPO1*, *KIF1C* (Fig. 2), or *XPO1* and *NFKBIA* (Fig. 3). Following mounting on the appropriate support, the samples were installed on the fluidic system. For the 20-round experiments, a series of 20 alternating hybridizations and stripping cycles for *XPO1* and *KIF1C* was performed. This enabled monitoring of signal quality over time, observing probe dehybridization, and tracking its progression throughout the

experiment. For the SABER experiments, the experiments were performed as described above.

## Mouse lung tissue

Studies were performed in accordance with the European Community's (2010/63/UE) recommendations for the care and use of laboratory animals. Experimental procedures were specifically approved by the ethics committee of the Institut Curie, CEEA-IC #118 (Authorization number APA-FIS#5479-201605271 0291841, given by the National Authority), in compliance with international guidelines. Ten to sixteen-week-old C57Bl6/J female mice were housed in the Institut Curie animal facility in compliance with European ethical procedures. Before tissue processing, mice were euthanized by cervical dislocation. To validate the hybridization of several genes on a section of fixed and cleared tissue, autoFISH was performed on mouse lung tissue. Briefly, mouse lungs were inflated with cold 4% PFA, fixed overnight at 4 °C, cryopreserved in 30% sucrose, and embedded in optimal cutting temperature (OCT) compound before sectioning[26].

## Analysis: sharpness measurements

3D DAPI image stacks were acquired and projected along the z-axis using maximum intensity projection. Nuclei were segmented from the projected images with CellPose[10]. For each nucleus, we determined its bounding box and computed the local contrast of every z-plane in the 3D images using Helmli's and Scherer local contrast algorithm[3,36] (Supplementary Note 1). For each nucleus, the plane with the highest focal contrast was selected for subsequent analysis. Composite images were then created by including the optimal focal-plane image for each nuclear mask, and background pixels were filled with the median z-index across all cells. Finally, we constructed box plots summarizing the extracted focus measurements for all nuclei in each condition.

## Analysis: sequential smFISH

Raw images are first reformatted into individual Z-stack.tif files, each corresponding to a field of view and hybridization round. To account for mechanical drift that may occur over successive rounds, we perform spatial registration using the SimpleITK package[37]. This process uses image-similarity metrics to align images across rounds, ensuring that RNA spots and cellular structures remain spatially aligned throughout the experiment.

For analysis of cell culture experiments, spots were detected using Big-FISH[8]. Big-FISH enhances punctate signals within the images using a Laplacian-of-Gaussian (LoG) filter, followed by the identification of RNA spots as local maxima. For lung tissue experiments, spot detection was performed using the recent deep-learning approach U-FISH[9]. U-FISH replaces Big-FISH's LoG filter with a deep-learning U-Net signal amplifier to enhance the true RNA spots signal and suppresses artefact signals. This U-Net feature is particularly advantageous for complex tissues such as the lung. Spot quality metrics, such as signal-to-noise ratio, background, and spot intensity, were computed using Big-FISH[8]. Nuclei segmentation on DAPI and cell segmentation, when cell staining was present, were performed with Cellpose[38]. Finally, all fields of view were stitched using the stitching ImageJ plugin[39]. The final step is to generate the spatially resolved gene-by-cell expression matrix from the previously localized RNA spots. However, because cell boundary staining is not always available in our experiments, the cell expression matrix is computed using our recently published point cloud segmentation method, ComSeg[33]. To perform cell type calling, we calculated the cosine distance between the cell expression vectors and the median centroids of each cell type, as defined in the reference scRNA-seq data from ref. 26. These gene markers allow for the classification of five distinct cell types: Alveolar Macrophage (AM) with *Chil3*, Alveolar Type 1 (AT1) with *Rtkn2*, Alveolar Type 2 (AT2) with *Lamp3*, and Epithelial Cells with *Pecam1 and Ptprb*. Cells are assigned to their nearest cell type cluster based on cosine distance. Given the limitations of our gene markers, not every cell can be classified. Therefore, we only classify cells with a cosine distance to the nearest cell-type centroid below 0.8 and with more than 10

detected RNAs. The analysis code is available at https://github.com/fish-quant/autofish-analysis.

## Analysis: image registration and RNA colocalization for SABER

To align image sets across rounds, in the absence of a dedicated fiducial marker, we generated binary spot maps and used phase cross-correlation to compute and apply optimal global shifts ($\Delta X$, $\Delta Y$). This shift is then applied to the spot coordinates of rounds (*XPO1*; 3 and 7) and (*NFKBIA*; 5 and 8) to precisely align them with their respective reference rounds 1 and 2. We then performed a secondary manual alignment using a Napari plugin[40] between *NFKBIA* (round 2) and the primary reference *XPO1* (round 1). The resulting translation is then applied to all rounds of *NFKBIA* (2, 5, and 8). Finally, spots were matched across rounds using the Hungarian algorithm to minimize inter-round distances. Colocalization was defined as a separation of <2 pixels ($\approx$2 µm) and reported as the percentage of total spots identified in that round.

## Statistics and Reproducibility

Consistent with proof-of-concept technology-development studies, validation relied on extensive technical sampling rather than biological replication. The statistical unit (n) corresponds to individual RNA molecules/spots, nested within cells and fields of view (FOVs). For each detected RNA, quantitative metrics including spot intensity, signal-to-noise ratio (SNR), local background, and co-localization scores (when applicable) were computed. Specific n values for cells, RNA counts, and per-molecule measurements are provided in the figure legends. Data are reported as Median ± SD, distributions, or boxplots (median, interquartile range, and 1.5× IQR whiskers), as indicated. No data were excluded, and all representative images reflect results consistently observed across all acquired FOVs.

## Reporting summary

Further information on research design is available in the Nature Portfolio Reporting Summary linked to this article.

## Data availability

The numerical source data underlying all graphs and charts in this study are publicly available in the Zenodo repository at https://doi.org/10.5281/zenodo.19047523. These data are organized by figure and panel as individual CSV files. The raw image data supporting the findings of this study are available from the corresponding author upon reasonable request due to their large file size.

## Code availability

Code to control the fluidic system is available at https://github.com/fish-quant/autofish. Code used for the analysis is available at https://github.com/fish-quant/autofish-analysis.

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

## Acknowledgements

This work has received financial support through the Institut Pasteur (Single Cell Seed Grant), Agence Nationale de la Recherche (ANR) grants LUSTRA (ANR-19-CE14-0015-03) and TRANSFACT (ANR-19-CE12-0007-02), as well as (ANR-24-INBS-0005 FBI BIOGEN). Further support from by a government grant managed by the Agence Nationale de la Recherche under the France 2030 program, with the reference numbers ANR-24-EXCI-0001, ANR-24-EXCI-0002, ANR-24-EXCI-0003, ANR-24-EXCI-0004, ANR-24-EXCI-0005. T.D, F.M. and C.W. acknowledge funding by Institut Pasteur. A.M. has received a PhD fellowship from the European Union's Horizon 2020 research and innovation program under the Marie Skłodowska-Curie grant agreement No 847718. H.L. is the recipient of a PhD fellowship from the International Student program from Paris-Saclay University. We would also like to thank Elric Esposito and Juliene Fernandez from the Photonic Biolmaging platform (PB) at the Pasteur Institute for their help setting up the imaging experiments. We thank the reviewers for their constructive and detailed feedback. Their suggestions have helped us clarify several points and improve the manuscript's overall quality.

## Author contributions

C.W. performed wet-lab experiments and contributed to the development of the fluidic system. T.D. and J.B. performed data analysis. M.L. provided microscopy expertise and support. C.S. contributed to the development of the fluidic system. H.L. and A.M. provided mouse samples. M.I.G. contributed to the writing of the manuscript. J.-A.L.-V. and C.F. provided technical insights and supervised the mouse-related work. T.W. contributed to data analysis. F.M. conceived the study, implemented the fluidic system, and led the writing and coordination of the manuscript. All authors reviewed and approved the final version of the manuscript.

## Competing interests

The authors declare no competing interests.
