## [Transparent Peer Review file · Communications Biology]

autoFISH: a modular toolbox for sequential single-molecule RNA FISH experiments

Corresponding Author: Dr Florian Mueller

Version 0:

Reviewer comments:

Reviewer #1

(Remarks to the Author)

In the manuscript by Weber et al. titled "autoFISH – a modular toolbox for sequential smFISH experiments," the authors present a comprehensive workflow for implementing custom-built fluidic device-assisted sequential fluorescence in situ hybridization (smFISH) experiments. Their goal is to make this approach more accessible and practical for the research community. The authors provide an overview of the workflow along with detailed, publicly available manuals, Python-based open-source code, and supporting materials compiled in a GitHub repository. These resources outline the design, construction, and operation of the fluidic instrumentation developed to perform multiple rounds of in situ hybridizations using a microscope.

As a proof of concept, the authors performed sequential smFISH on cultured HeLa cells and cleared mouse lung tissues. Their results demonstrate the utility of the autoFISH workflow. This work is significant because microfluidic devices are central to sequential smFISH and various multiplexed smFISH approaches, yet these devices remain a major bottleneck limiting broader accessibility. While fluidic device-assisted smFISH itself may not be novel—acknowledged by the authors as being "based on several previous studies"—this study shows that their affordable and reliable in-house approach fulfills the goal of enabling sequential smFISH experiments. Moreover, the authors emphasize that this work may inspire the research community to collaboratively optimize fluidic device-assisted protocols and expand instrumental modules for sequential smFISH.

Minor Concerns and Suggestions:

1. The manuscript covers a wide range of topics and includes valuable information. However, the results section, particularly the mouse lung tissue data, could be explained in more detail. For instance, if the goal was to showcase the autoFISH system's versatility in generating smFISH data after multiple rounds of sequential probing and stripping on cleared tissues, it would be helpful to emphasize methods for assessing RNA signal detectability. Additionally, a discussion of how detected RNA information could be applied to downstream analyses, such as clustering and cell-type designation, would strengthen the presentation.
2. In the "smFISH readout and stripping oligos" section of the Methods, the hybridization protocols could be clarified. The terminology for two different protocols (the base protocol and the SABER-related protocol) is somewhat mixed, which may confuse readers. Clear definitions of the sequence types (described as segments in the text) and the oligo probes designed to target these segments would make it easier to understand how each protocol operates.
3. In Figures 2 and 3, including sample sizes (e.g., number of replicates), error bars (e.g., standard deviation or standard error), and results from statistical tests (e.g., ANOVA) would make the data presentation more robust and complete.
4. In Figures 3C and 3D, the SABER method appears to amplify nuclear Kif1c signals while decreasing total RNA counts per cell. This observation raises the question of whether nonspecific binding and sequestration of secondary imaging probes to nuclear structures, such as chromatin, could be responsible.
5. In Figure 4, the finding that epoxide improves the sharpness of DAPI signals is intriguing. Quantifying this improvement—for example, using sharpness scores estimated in BIG-FISH Python—could provide a more objective assessment of the enhancement.

Reviewer #2

(Remarks to the Author)

Review manuscript #COMMSBIO-24-7622

The manuscript by Weber, Defard et al., titled “autoFISH - a Modular Toolbox for Sequential smFISH Experiments”, presents a comprehensive toolbox for automated single RNA molecule fluorescence in situ hybridization (smFISH) in mammalian cell lines and tissues. It includes detailed protocols, custom code for instrument control, and pipelines for spot quantification. This work is a great resource, offering a wealth of information to both novice and expert users wanting to implement smFISH experiments. It also includes validation experiments in mammalian cell lines and tissue samples. While the content is good, the manuscript text and figures require substantial revisions to increase clarity, remove typos, improve references etc. Furthermore, we recommend adding control analysis for the validation experiments. Below, we compiled a list of recommended changes. This peer-review was performed by the PI, as well as by a lab member expert in microfluidic systems, smFISH imaging and analysis.

Major points

- The text needs some rewriting. References are sparse and reviews are cited instead of primary literature. For instance, Pichon et al. 2018 is cited at multiple spots. This is a good review but is already outdated. We recommend prioritizing citing the primary literature. The introduction section feels unbalanced: a lot of emphasis is given to multiplexed methods, but then the SABER method, which is used in this manuscript, is just mentioned.
- A general comment about the figures: Some of the figures contain panels that seem distorted and stretched out, possibly because the images contain non-vectorized images. For instance, Figure 2C, where due to this distortion some of the median (or mean?) lines seem to have disappeared.
- Another general comment is that the Figure titles are not very informative, as they do not summarize the main results of each Figure. For example, the title of Figure 2 (“2. 20 Hybridisation rounds with autoFISH”) could be changed to a more informative title, such as “During 20 Hybridisation rounds alternating between the detection of XPO1 and KIF1C, autoFISH demonstrates efficient probe stripping and consistent mRNA detection”. Similarly, more informative titles could be added to the other figures.
- The figure legends should also include more information. The type of statistical analysis shown in the panels, the number of cells and biological replicates analysed in Figure 2, 3 and 4.
- The autoFISH GitHub is easily installed and a beautiful user interface was created that allows easy control of the components. However, we could not validate whether the code works as we are missing the necessary hardware components. The GitHub page also contains useful links to both the FISH protocols and setting up the microfluidic system. However, direct references in the Methods and Results to specific protocol files would be useful. As there are many files in the Google drive folder, this would make it easier to find the relevant information while reading the text.
- The protocols are now stored on Google drive. However, we suggest using repositories such as Zenodo or Mendeley for long-term storage. Both data storage services allow ‘doi’ generation and versioning of the documents.
- The autoFISH-analysis GitHub only installs on Linux machines which should be more clearly specified either in the text or on the GitHub page. For many users it would be convenient if a Windows install also existed (or a docker container).
- In the results section, it is not immediately clear which experiments were performed in the FSC2 chamber, and which were performed in the IBIDI 6-channel slides.
- Figure 2B is aimed at demonstrating stripping efficiency. To be more convincing, we recommend the following changes. First, in the figure, the labelling of the panels should be improved. The colour of the outlines of the pictures for Round 1 and 2 should be the same as the one of the XPO1 and KIF1 mRNAs, respectively. The merged image, with the yellow border, should be labelled with ‘Merge’, or a similar label. The three images are also missing a scale bar. Secondly, a quantification of the stripping efficiency should be presented in a separate panel. Not just a picture, but the co-localization analysis across many cells, RNAs and Rounds should be provided.
- Figure 2C. The figure legend does not describe whether the mean, or median is being shown in the image or what the whisker and outlier symbols indicate precisely (‘CI’ %, SD etc).
- Figure 3. To quantify the FISH efficiency using SABER FISH, you should quantify the co-localization of for XPO1 and KIF1C RNAs between smFISH and SABER rounds.
- Figures 3C-D. For KIF1C, you show rounds 7, and 6, while you quantify rounds 2,5,8,10. Please be consistent. Show pictures that are consistent with the quantifications.
- Figure 3D-E. For the KIF1C RNA, the number of RNAs detected during SABER FISH seems to be consistently lower than the smFISH counts. Is it possible that the increased spot sizes, due to SABER FISH amplification, and the localisation of this RNA to cell protrusions lead to the merging of neighbouring spots, and consequently some under-detection? It seems from the figure that the effect is minimal, but we think is important to discuss this result in the text. Also, in panel E you show that the SABER amplification leads to an increase of signal that is about 2-3 folds compared to the standard smFISH protocol. As the expected amplification is 10x, could you please comment on this result in the text?
- Figure 4B. The spots are not easy to see due to the size of the images and the cellular background. Could you improve the quality of the images, perhaps show the filtered data?
- Figure 4D. The figure legend is too vague. It says, “The clustering is based on the reference scRNAseq data”. However, from this description alone it is unclear what type of clustering of the FISH data was performed here and how it correlates to the scRNAseq data. This is better described in the main text, but some more explanation is needed here as well.

Minor points:

Below, we compiled a list of typos we found, that there may be others, especially in the protocols on Google drive. Please, proofread the text carefully before resubmission.

- Figure 1: There is a typo in the Figure legend after 'E' it says 'Aprovided'.
- Figure 2: There is a typo in the Panel E figure legend text saying "Pourcentage".
- In Figure 2, XPO1 and KIF1C are described in the figure legend as being placed right and left respectively in panel A. However, they seem to be placed left and right instead.
- Figure 2B: There is a typo in the legend. Didn't you merge Round 1 and Round 2 (not 20)?
- Figure 2G. The figure subtitle seems to contain a double space between 'Background' and 'intensity'.
- Figure 2H. The figure subtitle seems to be aligned to the left of the panel instead of the panel's middle. Also, the right axis border seems to be missing.
- Figure 2F-H are missing a y-axis label.
- Figure 3. In this figure you the label for the RNAs are lower case, while in the text and other figures are upper case. Please, be consistent.
- Figure 3A. The Marker sizes in the gel picture are missing. Additionally, it's unclear from the figure legend what type of gel and percentage was used.
- Figure 4. The names of the RNAs are lower case instead of upper case.
- In the Methods, you miss information about the microscope used to acquire the images shown in Figures 2-4.
- Figure 4C. Between the yellow cropped images, the triangles linking the images are not at the same height.
- Page 10, line 4. It seems that a space is missing here '3' _and'.
- Page 11, line 15. You wrote 'decremental'. Did you mean detrimental?

Reviewer #3

(Remarks to the Author)

-

Version 1:

Reviewer comments:

Reviewer #1

(Remarks to the Author)

In the manuscript by Weber et al. titled, "autoFISH: a modular toolbox for sequential smFISH experiments," the authors made a great effort to lay out a comprehensive workflow for implementing a custom-built fluidic device-assisted sequential fluorescence in situ hybridization (smFISH) experiments with the goal of making the very approach more accessible and available for the research community. In this work, they provided a manual with the critical points and tips for the designing, building and operating a sequential smFISH experiment. Furthermore, they made detailed manuals publicly available through the Github repository. As a proof of principle, the authors performed an exemplary sequential smFISH on cultured HeLa cells and cleared mouse lung tissues, and the results of which demonstrated impressive robustness and utility of the autoFISH workflow.

As I pointed out previously, this work is inspiring in that it provides service to broader research community and carries great values as a manual that may further inspire the research community to join the collective effort in optimizing fluidic device-assisted sequential smFISH protocols and the expansion of instrumental modules. The revised manuscript has appropriately addressed all points I made and further improved the overall quality of the manuscript substantially. Therefore, I recommend its publication.

Reviewer #2

(Remarks to the Author)

Review Paper # COMMSBIO-24-7622A

The manuscript presents autoFISH, a modular, open-source platform for automated sequential smFISH. The system integrates customizable hardware, flexible Python-based control software, and experimental protocols supporting both signal amplification (SABER) and tissue clearing. The authors validate the platform in HeLa cells and cleared mouse lung tissue, demonstrating robust signal detection, efficient probe stripping across 20 hybridization rounds, and improved nuclear signal retention using a GMA-based clearing modification. Overall, this work provides a cost-effective and adaptable alternative to commercial spatial transcriptomics systems.

In the revised version, the authors have addressed all reviewer comments, substantially improved the clarity of the manuscript, and included the requested experimental controls. We therefore support publication, pending the following minor revisions:

- Introduction: Key references to multiplexed FISH approaches are still missing (besides your own publication) after the sentence at line 3 ("Various adaptations of this method...").
- Methods (Microscopy): The filter sets used for image acquisition should be specified.
- Methods (smFISH readout and stripping oligos): The authors describe a strategy to test the quality of readout sequences ("We observed that some of the original readout sequences yielded lower signals during RNA smFISH..."). As this is a

valuable control, we suggest adding a schematic to the Supplementary Information to illustrate the strategy, along with representative image examples of both poor and robust readout sequences.

- Figure 2: The figure legends for panels C, F, G, and H do not describe what the lighter background shading represents in the line plots.
- Figure 3: The figure legend does not specify whether the central line in the boxplots represents the median or the mean, nor does it define the boxes and whiskers.
- Supplementary Figures 1 and 2: Scale bars are missing from the images and should be added.

We sincerely apologize for the extended delay in revising the manuscript. During this time, we faced significant challenges, including the passing of our lab chief, which required the last author to assume leadership responsibilities and significantly slowed progress. Additionally, we had to replace our microscope, and the new system experienced prolonged technical downtime.

We thank the reviewers for their constructive feedback and provide a point-by-point response below. Significant changes in the manuscript are underlined for clarity.

Reviewer #1

In the manuscript by Weber et al. titled “autoFISH – a modular toolbox for sequential smFISH experiments,” the authors present a comprehensive workflow for implementing custom-built fluidic device-assisted sequential fluorescence in situ hybridization (smFISH) experiments. Their goal is to make this approach more accessible and practical for the research community. The authors provide an overview of the workflow along with detailed, publicly available manuals, Python-based open-source code, and supporting materials compiled in a GitHub repository. These resources outline the design, construction, and operation of the fluidic instrumentation developed to perform multiple rounds of in situ hybridizations using a microscope.

As a proof of concept, the authors performed sequential smFISH on cultured HeLa cells and cleared mouse lung tissues. Their results demonstrate the utility of the autoFISH workflow. This work is significant because microfluidic devices are central to sequential smFISH and various multiplexed smFISH approaches, yet these devices remain a major bottleneck limiting broader accessibility. While fluidic device-assisted smFISH itself may not be novel—acknowledged by the authors as being “based on several previous studies”—this study shows that their affordable and reliable in-house approach fulfills the goal of enabling sequential smFISH experiments. Moreover, the authors emphasize that this work may inspire the research community to collaboratively optimize fluidic device-assisted protocols and expand instrumental modules for sequential smFISH.

Minor Concerns and Suggestions:

R1.1. The manuscript covers a wide range of topics and includes valuable information. However, the results section, particularly the mouse lung tissue data, could be explained in more detail. For instance, if the goal was to showcase the autoFISH system's versatility in generating smFISH data after multiple rounds of sequential probing and stripping on cleared tissues, it would be helpful to emphasize methods for *assessing RNA signal detectability*. Additionally, a discussion of how detected RNA information could be applied to *downstream analyses*, such as clustering and cell-type designation, would strengthen the presentation.

In response to reviewer feedback, we have incorporated a dedicated section in the Results describing our methodologies for assessing RNA signal detectability (page 10). Additionally, we

have expanded the Introduction to include relevant context on downstream analytical approaches (page 2).

R1.2. In the "smFISH readout and stripping oligos" section of the Methods, the hybridization protocols could be clarified. The terminology for two different protocols (the base protocol and the SABER-related protocol) is somewhat mixed, which may confuse readers. Clear definitions of the sequence types (described as segments in the text) and the oligo probes designed to target these segments would make it easier to understand how each protocol operates.

Thank you for the helpful comment. We agree that the explanations and nomenclature were unclear and have revised the relevant sections accordingly. Additionally, we added a new panel to Figure 1 to illustrate the different oligo types.

R1.3. In Figures 2 and 3, including sample sizes (e.g., number of replicates), error bars (e.g., standard deviation or standard error), and results from statistical tests (e.g., ANOVA) would make the data presentation more robust and complete.

We have revised the figures following the reviewers' recommendations. Sample sizes have been added to the figure legends, and standard deviations are now included in Figure 2C. For Figure 3, we incorporated new experimental data and added p-values to support statistical interpretation. Additionally, for Figure 4, we included a description of the cell clustering methodology.

R1.4. In Figures 3C and 3D, the SABER method appears to *amplify nuclear Kif1c signals* while decreasing total RNA counts per cell. This observation raises the question of whether nonspecific binding and sequestration of secondary imaging probes to nuclear structures, such as chromatin, could be responsible.

*We thank the reviewer for these observations. To clarify this point for the reviewers' attention, we revisited the oligos used in this experiment. We subjected them to blastn, retaining only oligos with Blast results showing two distinct groups of E values (typically $e < 10e-5$ for full-length matches and $e > 0$ for off-targets). **Using this criterion, we removed 20 of the 70 oligos initially designed.** We then compared the old and new oligo sets (see figure below). Images are shown with the same intensity scaling, and the background in the nucleus (blue outlines) is noticeably reduced, indicating that the background arises from non-specific binding of the primary oligos. Individual RNAs are slightly dimmer with the new set due to the reduced number of oligos, but are clearly visible and show the exact overall localization, including the cellular protrusions*

(orange arrows).

The reviewer also noted a slight reduction in KIF1C RNA levels for the SABER FISH experiments. As Reviewer 2 suggested, this may result from dense localization in cellular protrusions, where crowding can hinder RNA detection.

To permit a more **precise analysis of SABER usage in autoFISH**, we repeated the experiments using NFKB1a, a gene with a more diffuse localization pattern. These results are now presented in the new main Figure 3, and indicate efficient signal amplification and stripping.

R1.5. In Figure 4, the finding that epoxide improves the sharpness of DAPI signals is intriguing. Quantifying this improvement—for example, using sharpness scores estimated in BIG-FISH Python—could provide a more objective assessment of the enhancement. adf

We thank the reviewer for their valuable comments and suggestions. In response, additional data and analyses have been incorporated into the manuscript (**Main Text and Supplementary Figure 2**). A quantitative sharpness metric [1] was applied to evaluate image quality. This metric, based on local contrast, estimates focus by comparing intensity variations within a neighborhood: regions with higher local contrast are considered sharper, while areas with lower contrast are considered blurred. By computing this measure across the z-stack for each segmented nucleus, the optimal focal plane for each cell was identified. Using these optimal planes, composite images were generated to illustrate more clearly the positive effect of epoxide on DAPI signal, alongside quantitative measurements.

[1] Helmlí, S., Scherer, Adaptive shape from focus with an error estimation in light microscopy, Proc. Int. Symp. Image and Signal Processing and Analysis, 2001, pp. 188–193.

Reviewer #2 (Remarks to the Author)

The manuscript by Weber, Defard et al., titled “autoFISH - a Modular Toolbox for Sequential smFISH Experiments”, presents a comprehensive toolbox for automated single RNA molecule fluorescence in situ hybridization (smFISH) in mammalian cell lines and tissues. It includes detailed protocols, custom code for instrument control, and pipelines for spot quantification. This work is a great resource, offering a wealth of information to both novice and expert users wanting to implement smFISH experiments. It also includes validation experiments in mammalian cell lines and tissue samples. While the content is good, the manuscript text and figures require substantial revisions to increase clarity, remove typos, improve references etc. Furthermore, we recommend adding control analysis for the validation experiments. Below, we compiled a list of recommended changes. This peer-review was performed by the PI, as well as by a lab member expert in microfluidic systems, smFISH imaging and analysis.

Major points

R2.1 The text needs some rewriting. References are sparse and reviews are cited instead of primary literature. For instance, Pichon et al. 2018 is cited at multiple spots. This is a good review but is already outdated. We recommend prioritizing citing the primary literature. The introduction section feels unbalanced: a lot of emphasis is given to multiplexed methods, but then the SABER method, which is used in this manuscript, is just mentioned.

We have now included citations to key seminal papers and revised the introduction to provide a more balanced overview of multiplexed detection and signal amplification strategies. In response to Reviewer 1 (R1.1), we also added a new discussion section highlighting potential downstream analysis examples.

R2.2 A general comment about the figures: Some of the figures contain panels that seem distorted and stretched out, possibly because the images contain non-vectorized images. For instance, Figure 2C, where due to this distortion some of the median (or mean?) lines seem to have disappeared.

We revised the figures and addressed the identified issues.

R2.3 Another general comment is that the Figure titles are not very informative, as they do not summarize the main results of each Figure. For example, the title of Figure 2 (“2. 20 Hybridisation rounds with autoFISH”) could be changed to a more informative title, such as “During 20 Hybridisation rounds alternating between the detection of XPO1 and KIF1C, autoFISH demonstrates efficient probe stripping and consistent mRNA detection”. Similarly, more informative titles could be added to the other figures.

As suggested, we updated the figure title to make it more informative.

R2.4 The figure legends should also include more information. The type of statistical analysis shown in the panels, the number of cells and biological replicates analysed in Figure 2, 3 and 4.

We now provide more statistical information about the data shown in the figure legends (see also comment R1.3).

R2.5 The autoFISH GitHub is easily installed and a beautiful user interface was created that allows easy control of the components. However, we could not validate whether the code works as we are missing the necessary hardware components. The GitHub page also contains useful links to both the FISH protocols and setting up the microfluidic system. However, direct references in the Methods and Results to specific protocol files would be useful. As there are many files in the Google drive folder, this would make it easier to find the relevant information while reading the text.

We thank the reviewer for the positive comment. As recommended, we provide all protocols on Zenodo (see R2.6).

R2.6 The protocols are now stored on Google drive. However, we suggest using repositories such as Zenodo or Mendeley for long-term storage. Both data storage services allow 'doi' generation and versioning of the documents.

We thank the reviewer for this suggestion. We now provide all protocols on Zenodo <https://zenodo.org/records/17965897>

R2.7 The autoFISH-analysis GitHub only installs on Linux machines which should be more clearly specified either in the text or on the GitHub page. For many users it would be convenient if a Windows install also existed (or a docker container).

We thank the reviewer for pointing out the issues with the installation. We have added a requirements.txt file, allowing for a one-liner installation (pip install -r requirements.txt) on Linux, Windows, and macOS. We believe these additions eliminate the Linux-only limitation highlighted by the reviewer while maintaining a lightweight and transparent installation procedure.

R2.8 In the results section, it is not immediately clear which experiments were performed in the FSC2 chamber, and which were performed in the IBIDI 6-channel slides.

Due to inconsistent results when repeating the experiments using the Ibidi system, we decided to omit its mention from this manuscript. We are investigating this further and to include it in future publications.

R2.9 Figure 2B is aimed at demonstrating stripping efficiency. To be more convincing, we recommend the following changes. First, in the figure, the labelling of the panels should be improved. The colour of the outlines of the pictures for Round 1 and 2 should be the same as the one of the XPO1 and KIF1 mRNAs, respectively. The merged image, with the yellow border, should be labelled with 'Merge', or a similar label. The three images are also missing a scale bar. Secondly, a quantification of the stripping efficiency should be presented in a separate panel. Not just a picture, but the co-localization analysis across many cells, RNAs and Rounds should be provided.

We thank the reviewer for the suggestion and have updated the figure accordingly. We also performed the recommended co-localization analysis (Figure 2E), which shows strong self-co-localization of XPO1 and KIF1C across rounds and minimal cross-co-localization, providing quantitative support for effective probe stripping.

R2.10 Figure 2C. The figure legend does not describe whether the mean, or median is being shown in the image or what the whisker and outlier symbols indicate precisely ('CI' %, SD etc).

We have now included the 50% percentile interval in the figure.

R2.11 Figure 3. To quantify the FISH efficiency using SABER FISH, you should quantify the co-localization of for XPO1 and KIF1C RNAs between smFISH and SABER rounds.

We added new SABER data (Figure 3 and R1.4) and included a co-localization analysis confirming efficient probe stripping.

R2.12 Figures 3C-D. For KIF1C, you show rounds 7, and 6, while you quantify rounds 2,5,8,10. Please be consistent. Show pictures that are consistent with the quantifications.

We corrected these errors for the new figure.

R2.13 Figure 3D-E. For the KIF1C RNA, the number of RNAs detected during SABER FISH seems to be consistently lower than the smFISH counts. Is it possible that the increased spot sizes, due to SABER FISH amplification, and the localisation of this RNA to cell protrusions lead to the merging of neighbouring spots, and consequently some under-detection? It seems from the figure that the effect is minimal, but we think it is important to discuss this result in the text. Also, in panel E you show that the SABER amplification leads to an increase of signal that is about 2-3 folds compared to the standard smFISH protocol. As the expected amplification is 10x, could you please comment on this result in the text?

As also noted by Reviewer 1 (R1.4), KIF1C RNA levels appeared slightly reduced in the SABER runs. We agree with reviewer 2's suggestion that this may result from strong local enrichment of KIF1C in protrusions. To enable more accurate quantification, we repeated the experiments using a gene with a more diffuse expression pattern (R1.4).

The reviewer raised an important point regarding SABER signal amplification and an apparent inconsistency in our original data. We realized that there is one main explanation for this difference: in our regular smFISH experiments, imaging oligos are labeled at both the 3' and 5' ends, whereas SABER imaging oligos are labeled only at the 5' end. This reduces the expected signal amplification between our smFISH and the original SABER with 8-10 binding sites to a factor of 4-5, closer to our reported value of 3.

*To further investigate this effect, we conducted additional experiments (**Supplementary Figure 1**). As a more controlled comparison, we used directly synthesized SABER repeats (IDT) with three binding sites. We compared them to SABER amplicons generated after 30 minutes of amplification, yielding 8-10 binding sites. We also tested imaging oligos carrying either one or two fluorophores. The results showed a consistent increase in signal from 3 to 8 binding sites. Interestingly, dual-fluorophore oligos also enhanced signal, though the effect was less pronounced with larger SABER amplicons, possibly due to fluorophore crowding or quenching.*

R2.14 Figure 4B. The spots are not easy to see due to the size of the images and the cellular background. Could you improve the quality of the images, perhaps show the filtered data?

We enhanced image contrast to improve the visibility of the smFISH signal for each gene.

R2.15 Figure 4D. The figure legend is too vague. It says, "The clustering is based on the reference scRNAseq data". However, from this description alone it is unclear what type of clustering of the FISH data was performed here and how it correlates to the scRNAseq data. This is better described in the main text, but some more explanation is needed here as well.

We added the missing information to the figure legend.

R2.16 Minor points:

Below, we compiled a list of typos we found, that there may be others, especially in the protocols on Google drive. Please, proofread the text carefully before resubmission.

- Figure 1: There is a typo in the Figure legend after 'E' it says 'Aprovided'.
- Figure 2: There is a typo in the Panel E figure legend text saying "Pourcentage".
- In Figure 2, XPO1 and KIF1C are described in the figure legend as being placed right and left respectively in panel A. However, they seem to be placed left and right instead.
- Figure 2B: There is a typo in the legend. Didn't you merge Round 1 and Round 2 (not 20)?
- Figure 2G. The figure subtitle seems to contain a double space between 'Background' and 'intensity'.
- Figure 2H. The figure subtitle seems to be aligned to the left of the panel instead of the panel's middle. Also, the right axis border seems to be missing.
- Figure 2F-H are missing a y-axis label.

- Figure 3. In this figure you the label for the RNAs are lower case, while in the text and other figures are upper case. Please, be consistent.
- Figure 3A. The Marker sizes in the gel picture are missing. Additionally, it's unclear from the figure legend what type of gel and percentage was used.
- Figure 4. The names of the RNAs are lower case instead of upper case.
- In the Methods, you miss information about the microscope used to acquire the images shown in Figures 2-4.
- Figure 4C. Between the yellow cropped images, the triangles linking the images are not at the same height.
- Page 10, line 4. It seems that a space is missing here '3'_and'.
- Page 11, line 15. You wrote 'decremental'. Did you mean detrimental?

Thank you for thoroughly reviewing the manuscript and protocol. We have corrected these typos and others and enhanced the wording throughout the manuscript.

We thank the reviewers for their constructive feedback and provide a point-by-point response below. Changes in the manuscript are underlined for clarity.

Reviewer #2

The manuscript presents autoFISH, a modular, open-source platform for automated sequential smFISH. The system integrates customizable hardware, flexible Python-based control software, and experimental protocols supporting both signal amplification (SABER) and tissue clearing. The authors validate the platform in HeLa cells and cleared mouse lung tissue, demonstrating robust signal detection, efficient probe stripping across 20 hybridization rounds, and improved nuclear signal retention using a GMA-based clearing modification. Overall, this work provides a cost-effective and adaptable alternative to commercial spatial transcriptomics systems.

In the revised version, the authors have addressed all reviewer comments, substantially improved the clarity of the manuscript, and included the requested experimental controls. We therefore support publication, pending the following minor revisions:

R2.1 Introduction: Key references to multiplexed FISH approaches are still missing (besides your own publication) after the sentence at line 3 (“Various adaptations of this method...”).

We added a more comprehensive list of references.

R2.2 Methods (Microscopy): The filter sets used for image acquisition should be specified.

We now provide the references for filter sets on our microscopes.

R2.3 Methods (smFISH readout and stripping oligos): The authors describe a strategy to test the quality of readout sequences (“We observed that some of the original readout sequences yielded lower signals during RNA smFISH...”). As this is a valuable control, we suggest adding a schematic to the Supplementary Information to illustrate the strategy, along with representative image examples of both poor and robust readout sequences.

We added Supplementary Figure 3 to the manuscript, which shows the schematic and representative example images.

R2.4 Figure 2: The figure legends for panels C, F, G, and H do not describe what the lighter background shading represents in the line plots.

We now provide this information in the figure legend.

R2.5 Figure 3: The figure legend does not specify whether the central line in the boxplots represents the median or the mean, nor does it define the boxes and whiskers.

We updated the figure legend with the missing information.

R2.6 Supplementary Figures 1 and 2: Scale bars are missing from the images and should be added.